# Combined Trans-Arterial Embolization and Ablation for the Treatment of Large (>3 cm) Liver Metastases: Review of the Literature

**DOI:** 10.3390/jcm11195576

**Published:** 2022-09-22

**Authors:** Eliodoro Faiella, Alessandro Calabrese, Domiziana Santucci, Carlo de Felice, Claudio Pusceddu, Davide Fior, Federico Fontana, Filippo Piacentino, Lorenzo Paolo Moramarco, Rosa Maria Muraca, Massimo Venturini

**Affiliations:** 1Department of Radiology, Sant’Anna Hospital, Via Ravona, 22042 San Fermo della Battaglia, Italy; 2Department of Radiological Sciences, Oncology and Pathology, Umberto I Hospital, Sapienza University of Rome, Viale del Policlinico 105, 00161 Rome, Italy; 3Unit of Computer Systems and Bioinformatics, Department of Engineering, Campus Bio-Medico University, Via Alvaro del Portillo 21, 00128 Rome, Italy; 4Regional Referral Center for Oncologic Disease, Department of Oncological and Interventional Radiology, Businco Hospital, A.O. Brotzu, 09100 Cagliari, Italy; 5Department of Diagnostic and Interventional Radiology, University of Insubria, Ospedale di Circolo e Fondazione Macchi, 21100 Varese, Italy; 6Diagnostic and Interventional Radiology Unit, ASST Settelaghi, 21100 Varese, Italy; 7Insubria University, 21100 Varese, Italy

**Keywords:** secondary liver lesions, interventional radiology, embolization

## Abstract

(1) Background: The aim of this review was to determine the state of clinical practice in the role of the combined approach of embolization and ablation in patients with secondary liver lesions greater than 3 cm who are not candidates for surgery, and to study its safety and efficacy. (2) Methods: Two reviewers conducted the literature search independently. Eight articles on the combined approach of embolization and ablation in secondary liver lesions were selected. (3) Results: The studies were published between 2009 and 2020. Two studies were prospective in design. The sample size was < 100 patients for all studies. All studies demonstrated the safety of the combined approach based on the low complication rate. Some studies lamented non-uniform systemic chemotherapy regimens and the variability in the sequence of embolization and ablation. (4) Conclusions: This review presents the combined approach of ablation and embolization in liver lesions greater than 3 cm as a safe therapeutic procedure with positive effects on patient survival. Prospective and multicentric studies are needed to further evaluate its efficacy.

## 1. Introduction

The liver is one of the principal sites for the spread of distant metastases, due to its rich vascularity and unique cellular architecture, making the organ a thriving ground for cancer cells. Of the approximately 2.4 million cancer patients in the SEER database from 2010 to 2015, about 5.1% presented liver metastasis at diagnosis [1,2]. Breast cancer is the most common primary tumor in women aged 20–50 with liver metastasis, while colon cancer is the most frequent in men aged 20–50, followed by rectum, lung, and neuroendocrine pancreas tumors. The 1-year survival of cancer patients with liver metastases is 15.1%, compared to 24.0% of those with non-hepatic metastases [2].

Resection is considered the management reference standard in patients with liver metastases from colorectal cancer, with five-year survival ranging from 25% to 44%, and an operative mortality of 0–6.6% [3]. Colorectal metastases are defined as resectable if a R0 resection can be performed, leaving at least 20–25% of the total liver volume with adequate inflow, outflow, and biliary drainage [4]. Unfortunately, only about 20% of patients with metastases are candidates for surgery, due to the anatomic limitations, number, location, and extent of liver lesions, insufficient liver function, and comorbidities [5].

Modern chemotherapy regimens, such as 5-fluorouracil/leucovorin with oxaliplatin or camptothecin and monoclonal antibodies, have increased the median survival of patients with unresectable liver metastases to 20–26 months, although with an associated risk of skin reactions and impaired liver function [6].

Alternatively to surgical treatment and systemic chemotherapy, interventional procedures such as percutaneous ablation and trans-arterial embolization (TAE) have recently emerged with the aim to reduce disease recurrence, increase the number of patients able to undergo hepatectomy by acting as a bridging therapy to resection, and prolong the survival of patients with liver metastases who are not candidates for surgery.

The most commonly used ablative approaches are radiofrequency ablation (RFA) and microwave ablation (MWA). The main goal of ablative therapy is complete ablation, similar to R0 resection, with a margin of at least 10 mm around the outer margin of the target lesion. Compared to RFA, MWA allows the achievement of a larger volume of cell necrosis in a shorter time and with higher temperatures. MWA is also less dependent on change in the morphology of the treated area due to heat sink effects from adjacent vasculature. On the other hand, larger areas of necrosis increase the risk of possible complications from collateral damage to non-target organs. Thus, MWA is currently considered superior to RFA for perivascular lesions and those larger than 3 cm, whereas RFA is safer for peribiliary lesions because of the less aggressive production of heat and greater predictivity of the ablation zone [7].

Trans-arterial embolization (TAE) and trans-arterial chemoembolization (TACE) exploit the predominantly arterial supply of hepatic neoplastic lesions, including hepatocellular carcinoma (HCC) and liver metastases. The combination of embolization particles and chemotherapeutics results in high-dose chemotherapy access to the liver and selective ischemia of neoplastic tissues (6). TACE has shown to be effective in the treatment of not only HCC and cholangiocarcinoma, but also of liver metastases from colorectal cancer [8], neuroendocrine tumors [9], and breast cancer [10], among others. Centers differ in the choice of chemotherapeutic drug, embolization agent, and retreatment schedule [6].

Recent literature has already demonstrated the benefits on the survival and recurrence-free life of patients with an HCC nodule larger than 5 cm or multiple HCC nodules larger than 3 cm, who underwent a combined approach (in either sequence) of ablation and TAE/TACE [11]. Less evidence exists regarding the possible use of this synergistic approach, as well as on the possible benefits and complications, when applied to liver metastases.

The purpose of this review was to determine the state of clinical practice in the role of the combined approach of TAE and ablation in patients with secondary liver lesions greater than 3 cm who are not candidates for surgery, and to study its safety and efficacy.

## 2. Materials and Methods

Databases, such as PubMed, Google Scholar, Scopus, and Web of Science, were employed for the research, using the following strings: [(“embolization” OR “embolisation” OR “chemoembolization”) AND (“ablation”) AND (metastatic OR metastases OR metastasis) AND (“liver” OR “hepatic”)]. All full studies related to the combined embolization and ablation approach on liver lesions greater than 3 cm published prior to August 2022 were included. We completed the search by manually reviewing the bibliography of all selected articles.

Two reviewers independently conducted the search, reviewed all abstracts, selected and retrieved studies, and extracted data from each study. The inclusion criteria were as follows: (a) clinical trials must involve patients with metastatic liver lesions treated with embolization combined with ablation; (b) studies including lesions with a diameter greater than 3 cm; (c) studies involving humans only; (d) articles accessible through our institution; and (e) English publications only. Articles were excluded if no lesion size was documented in the text. Case studies, abstracts, reviews, letters to the editor, editorials, and commentaries were excluded.

The following data were extracted from each study: title, authors, country of study, year and journal of publication, study design (retrospective or prospective), number and age of patients, maximum liver lesion diameter, primary tumor, type of treatment approach, median survival, and major complications.

The review was performed according to the guidelines of Preferred Reporting Items for Systematic Reviews and Meta-Analyses (PRISMA) [12]. A flow chart of the selection process is depicted in Figure 1.

## 3. Results

From a total of 853 articles relevant to the search, 9 full studies were considered suitable and then collected. Study characteristics, as recorded by the reviewers, are shown in Table 1 and Table 2. TAE/TACE methods are reported in Table 3.

All the studies were published between 2009 and 2022. Two studies (22%) were prospective in design. A mean of 36.77 ± 20.63 patients (range: 19–88) were studied. The mean age of the population of the studies was 63.48 ± 6.45 years (range: 55–74.1 years); all studies included both men and women, with the exception of Wang et al., who uniquely considered female patients with liver metastases from breast cancer [18]. The mean lesion size of the studies was 3.69 ± 0.86 cm (range: 2.2–4.7 cm); all studies included patients with both isolated liver lesions and multiple lesions. The mean follow-up period was 23.8 ± 9.12 months (range: 10.3–39.2 months).

Gadalata et al. [13] evaluated the efficacy and safety of a “single step” approach with TACE and RFA in 34 patients with 37 primary and 21 secondary liver neoplasms with unresectable lesions or who refused surgery. After a mean follow-up period of 14 months (range: 2–17), 31 patients with 51 nodules were evaluated for local response and 34 for toxicity. The first follow-up was performed by contrast-enhanced 48 h CT after treatment, then after 1 month, and then every 3 months. Complete necrosis was observed in 45/51 nodules, and local disease progression in 6/51. No correlation was found between local tumor progression and histopathology, nodule size, or treatment. The treatment was well-tolerated with moderate hepatic and hematologic toxicity: grade 3–4 thrombocytopenia occurred in 4/37 sessions in cirrhotic patients (10.8%), while grade 3–4 anemia and leukopenia were observed in two cases; six patients (18%) experienced long-lasting fatigue. One cirrhotic patient died due to the onset of acute liver failure.

Fong, Z.V. et al. [14] investigated the feasibility of the combined approach of intra-arterial embolization and ablation (RFA, MWA, or CA) by dividing 32 patients with unresectable liver metastases from colorectal adenocarcinoma into two groups: Group 1 (18 patients) was treated with embolization followed by ablation, while Group 2 (14 patients), with larger lesions, was treated with embolization followed by ablation. Twenty-seven patients underwent TACE, while five patients were treated with radioembolization with yttrium-90 resin microspheres (90Y). Ablation was performed within six weeks after intra-arterial treatment. The median follow-up period was 39.2 months; two- and five-year survival was 75.0% and 10.1%, respectively. No significant difference was found in the overall survival of the two groups, probably due to the effect of pretreatment with embolization. There was no treatment-related mortality. Morbidity included one liver abscess, two cases of postoperative ileus, one cholecystitis, one apical pneumothorax, and one portal vein thrombosis.

Kan et al. [15] studied 19 patients with liver metastasis, 12 from colorectal cancer, 5 from gastric cancer, and 2 from esophageal cancer, treated with TACE followed by RFA. Patients were evaluated every 1–3 months by blood tests and imaging examinations, including ultrasound, CT, and MRI. After a median follow-up period of 21.3 months, 10/19 patients (52.6%) had distal intrahepatic recurrences and/or extrahepatic secondary tumors, and 6/19 patients (31.2%) had local recurrence, defined as the appearance of a tumor within or on the periphery of the original treated lesion. Larger tumor size and hypovascular lesion enhancement were significantly associated with more frequent recurrence. The 1-, 2- and 3-year survival rates were 89.4%, 52.6%, and 35.1%, respectively. The median survival time for the group was 35.2 months. Two major complications were observed, all related to RFA (bile duct injury and segmental hepatic infarction).

Wu et al. [16] reviewed their center’s experience in the therapeutic efficacy and safety of the combined approach of MWA followed by TACE in the treatment of metastatic liver lesions from intestinal cancer. Of 30 patients with 43 lesions undergoing treatment, 20 had colon cancer and 10 had rectal cancer. The median follow-up time was 26.5 ± 10.4 months; post-treatment evaluation was performed with contrast-enhanced CT and/or MRI 1 month after the procedure, and every 6 to 7 weeks until the end of follow-up. Complete ablation was found in 35/43 (81.4%) lesions. One-month post-treatment complete response was observed in 8 (26.7%) patients and partial response in 17 (56.7%) patients, while stability of disease and disease progression were observed in 2 and 3 patients, respectively. The median overall survival of patients with tumor loads of 25%, 26–50%, and 50% of liver volume were 33.1 months, 14.7 months, and 6.1 months, respectively. Median progression-free survival and overall survival following the MWA-TACE procedure for the entire cohort were 5.0 months and 11.0 months, respectively, with 12-month and 24-month survival rates of 46.7% and 25.4%, respectively. Neither treatment-associated mortality nor significant postoperative complications were observed.

Yamakado et al. [17] investigated the therapeutic benefits of RFA combined with TACE (degradable starch microsphere and Mytomycin C) in 25 non-surgical candidate patients with 38 liver metastases from colorectal cancer. RFA was performed before TACE. Initial response to treatment was assessed after 3–5 days with contrast-enhanced CT. Follow-up CT imaging was performed 1 month after the procedure, and then every 3 months. MRI and PET were used in cases of suspected CT progression. A total of 11 patients (44%) underwent therapy once, and 14 patients (56%) twice due to multiple tumors or insufficient ablation margins. Complete response was achieved in all 38 tumors (100%). The mean follow-up period was of 34.9 ± 9.2 months: 15 patients developed recurrence of liver tumor. The 2-year overall and recurrence-free survival rates were 88% and 63.3%, respectively. Tumor size correlated significantly with the risk of local tumor progression. The CEA level was positive in 17 (68%) patients at the time of combination therapy, and became negative in 12 patients (70.6%) at 1 month and in 15 patients (88.2%) after 3 months. No treatment-related mortality was observed, while fever was the only adverse event, requiring treatments in 2 patients (8%).

Wang et al. [18] studied 88 patients with liver metastases from breast cancer. The study population was divided into a group of patients treated with combined TACE and RFA (50 patients) and a control group of patients treated with TACE alone (38 patients). After a median observation period of 20 months (range: 6–35 months), an increase in therapeutic effect rate, progression-free survival, median survival time, and survival rate were noted in the observation group compared to the control one. No significant differences in complication rate were found between the two groups.

Alexander, E.S. et al. [19] assessed the effects of a combined TACE and ablation (RFA, MWA, or CA) approach in 42 patients with metastatic liver lesions (18 colorectal cancers, 5 sarcomas, 5 cholangiocarcinomas, 4 pancreatic neuroendocrine tumors, 4 pancreatic adenocarcinomas, 3 breast cancers, and 3 others). Ablation was performed the day after TACE in 42/44 patients, and within 14 days in 2/44 patients. Forty-five percent of patients presented a complication, of which eight (9%) were of major grade. Tumor response evaluation according to mRECIST criteria was available for 39/42 patients and 41/44 lesions, with a median follow-up of 10.3 months: 32 lesions (78.0%) demonstrated a complete response, 8 lesions (19.5%) a partial response, and 1 lesion (2.4%) a progression of disease. A total of 14 out of 38 patients had local tumor recurrence, 29/38 had remote hepatic recurrence, and 26/38 had extrahepatic recurrence. Overall survival was 55% (95% CI 40–71%) at 1 year and 30% (95% CI 16–45%) at 2 years. Freedom from local recurrence was 61% at 1 year and 50% at 2 years. Tumor size was not a predictor of progression. Patients who received 90% of the targeted dose had a significantly higher rate of complete response compared to those patients who received > 90%. History of prior pancreaticoduodenectomy and of prior ablation were associated with a longer freedom from hepatic recurrence. Complications occurred in 19 patients (45%).

Faiella et al. [20] evaluated the safety and efficacy of a two-step single-session combined treatment with TAE and MWA on large (>3 cm) liver metastases in 22 patients with 24 total lesions (10 breast cancers, 10 colorectal cancers, 2 neuroendocrine tumors, and 2 leiomyosarcomas). Follow-up with CT scans was performed at 1, 3, 6, 12, and 24 months; technical success was achieved in all treatments (24/24), and no residue/recurrence was detected until the last follow-up. The study also compared the final volume obtained by the two techniques and the expected ablation volume of the stand-alone MWA, with an average increase of 196%. The average final volume obtained by the two techniques was four times larger than the average initial nodule volume. The major complication rate was 4% with only one case (1/24), the minor complication rate was 4%, and three patients developed minor side effects.

Kobe et al. [21] evaluated the safety and efficacy of TACE combined with percutaneous thermal ablation (MWA, RFA, or CA) in patients with liver metastases > 3 cm, in 39 patients with 46 total lesions (8 colorectal cancers, 10 sarcomas, 10 adrenal carcinomas, 3 thyroid carcinomas, 4 neuroendocrine tumors, 3 breast cancers, and 8 others). All procedures were performed in a single session. Follow-up with imaging was performed at 1, 3, 6, and 12 months, and then every 12 months thereafter. The mean follow-up period was 24 months (range: 2–113 months); 7/46 (15%) lesions developed local progression, and in 29/39 patients (74%), systemic disease progression was observed after a mean time of 9.8 months. The overall survival rates at 1 and 2 years were 95% and 77%, respectively. Four grade 2 complications were reported (4/39; 10%); no treatment-related deaths were observed.

## 4. Discussion

Non-operative management in patients with secondary liver lesions who are not candidates for surgery is a very current topic which has gained more attraction in the clinical field due to the progressive development of interventional radiology techniques. Although surgery remains the gold standard and still represents the only chance for a definitive cure, the ablation treatment, in combination with chemotherapy, could represent a bridge to surgical therapy and/or increase patients’ overall survival [22]. Several studies have already described the efficacy and safety of the combined approach of ablation and embolization in HCC nodules [11], but few studies exist in the literature that specifically study the combined approach for the management of secondary liver lesions.

All studies in this review confirmed the efficacy of combined ablation and embolization therapy. Considerable variation was found in the methodology of the combined approach, especially regarding the modality of ablation, drugs used in chemoembolization, and specific therapeutic sequence and timing between the two procedures. Some studies preferred to perform TAE as the first treatment, with the rationale of reducing the vascularization of larger tumors by reducing heat sink effects and increasing ablative phenomena. Most studies performed the two treatments in a single session, with the dual purpose of preventing further revascularization in the time between the two procedures, and reducing the hospital stay time and/or individual patient accesses to the hospital. This allows for improved compliance in otherwise frail patients, usually forced into repeated hospital admissions.

All studies demonstrated the safety of the combined approach based on the low complication rate. Treatment-related mortality was found in only one study, and complications related to embolization and ablation were infrequent and usually managed conservatively. Complication rates were no different than those of individual ablation and embolization methods, as defined by the CIRSE Standards of Practice on Hepatic Transarterial Chemoembolisation and Thermal Ablation [23,24]. Despite the overall safety, there was considerable variation among studies in the rate of major complications, which ranged from 0% to 19%. This was due to both the variability in the treatment choices mentioned above and in the study populations, which differed in pretreatment clinical characteristics and treatment history for both surgery and chemotherapy. Advanced age and larger tumor size were considered risk factors for mortality in RFA [25].

The main limitations of the studies were the relatively small population sample sizes (no study exceeded 100 patients), the monocentric nature of the studies, the selection bias introduced by the retrospective design, since only two studies were prospective, and the relatively short follow-up time considered. Some studies lamented non-uniform systemic chemotherapy regimens and the variability in the sequence of embolization and ablation, for which there is still no consensus. The study by Wang et al. [18] was the only study that performed a direct comparison of the therapeutic effects of an observational group treated with TACE and RFA and a control group treated with TACE alone, proving an increase in the effective rate, progression-free survival, median survival time, and survival rate of the observation group, as well as a complication rate that was not statistically significant between the two groups. All the other studies proved their efficacy by comparison with previous studies, some of them concerning the evaluation of the focused combination approach to HCC and not to secondary liver lesions. The overall survival was reported by only four works. However, there was a wide range among the studies: while Wu et al. [16] and Alexander et al. [19] reported a survival rate of 46.7% and 55% at 12 months, respectively, the other three works showed very high rates (89.4%, 88%, and 95%) [15,17,21]. The two works that analyzed the overall survival, comparing the group treated with the combined approach and a control group, reported, in both cases, better results for the observation group [14,18]. The data are also few about progression-free survival the data, with only two works reporting an absolute value.

We may conclude that the treatment with TACE and ablative techniques is an emerging and promising approach, and this review demonstrates the effectiveness and efficacy on a small set of samples. At the moment, there are no long-term results, and prospective studies are next to come. Based on the analysis of the results, we can come to the conclusion that one of the main factors in obtaining optimal procedural outcomes is adequate patient selection, as well as the availability of qualified and competent staff with access to specific devices within the referral center.

Our review itself has some limitations. Because this type of combined approach is of recent application to the clinical setting (8/9 of the studies are post-2010), a relatively low number of studies were analyzed. This relative actuality of the approach is reflected in the considerable variability among the studies regarding the population being examined and the specific combination therapy, as well as the presentation of survival and therapeutic efficacy data, preventing us from performing a direct and uniform comparison among studies and an effective statistical analysis that could appropriately describe the differences between the studies themselves, and for which reason we performed only a descriptive analysis. We could only evaluate two prospective observational studies, so the quality of the current evidence is further reduced. In addition, we eliminated papers in non-English languages or that were not accessible from our institution, reducing the evidence.

## 5. Conclusions

This review presents the combined approach of ablation and embolization in secondary liver lesions greater than 3 cm as a safe therapeutic procedure with positive effects on patient survival. More studies, particularly prospective in design, are needed to further evaluate its efficacy. Despite the variability in method, the fact that all studies proved the safety and therapeutic efficacy of this procedure is encouraging.

## Figures and Tables

**Figure 1 jcm-11-05576-f001:**
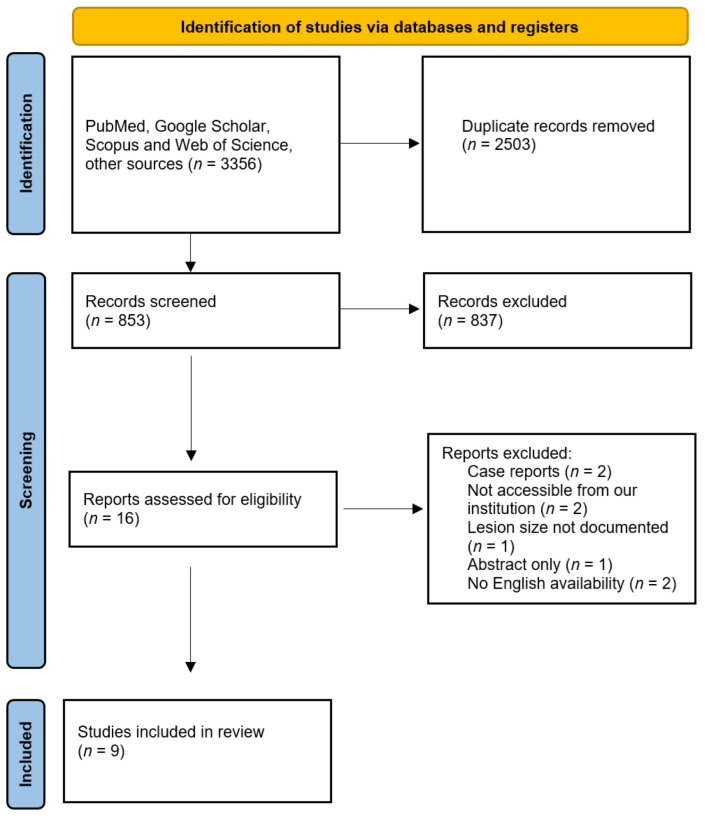
PRISMA flowchart showing inclusion and exclusion of the studies.

**Table 1 jcm-11-05576-t001:** Characteristics of the studies included in the review (part 1).

Authors	Publication Year	Journal	Country of Study	Study Design	Number of Patients	Age (Years)
Gadaleta et al. [13]	2009	In Vivo	Italy	Retrospective	34 (21 with HCC, 13 with liver metastases)	70 (range: 47–83)
Fong, Z.V. et al. [14]	2012	The American Surgeon	USA	Retrospective	32	74.1 (range: 50–96)
Kan et al. [15]	2016	Journal of Huazhong University of Science and Technology	China	Retrospective	19	61.8 (range: 34–82)
Wu et al. [16]	2016	OncoTargets and Therapy	China	Retrospective	30	61.6 (range: 44–78)
Yamakado et al. [17]	2017	CardioVascular and Interventional Radiology	Japan	Prospective	25	70.2 (range: 55–82)
Wang et al. [18]	2017	Oncology Letters	China	Prospective	88 (control group = 50; observational group = 38)	56.7
Alexander, E.S. et al. [19]	2018	Abdominal Radiology	USA	Retrospective	42	62 (range: 38–83)
Faiella et al. [20]	2020	International Journal of Hyperthermia	Italy	Retrospective	22	58.5 (range: 43–81)
Kobe et al. [21]	2022	Diagnostic and Interventional Imaging	France	Retrospective	39	55 (range: 28–77)

**Table 2 jcm-11-05576-t002:** Characteristics of the studies included in the review (part 2).

Authors	Average Maximum Lesion Diameter	Primary Tumor	Therapy	Median Survival	Major Complications
Gadaleta et al. [13]	2.5 cm (1–6 cm)	Colorectal 9/13 (69%) Breast 3/13 (23%) Ovarian 1/13 (8%)	TACE + RFA	N/A	1 died after acute liver failure
Fong, Z.V. et al. [14]	4.4 cm (1.7–7.9 cm)	Colorectal 32/32 (100%)	TACE and 90Y + MWA	36 months	1 liver abscess, 2 postoperative ileus, 1 cholecystitis, 1 apical pneumothorax, and 1 portal vein thrombosis
Kan et al. [15]	4.2 cm (1.5–7.8 cm)	Colorectal 12/19 (63%) Gastric 5/19 (26%) Esophagus 2/19 (11%)	TACE + RFA	35.2 months	1 died after bile duct injury, 1 segmental hepatic infarction
Wu et al. [16]	4.4 ± 2.6 cm (1.4–10.0 cm)	Colorectal 30/30 (100%)	TACE + MWA	11.0 months	None
Yamakado et al. [17]	2.2 ± 0.9 cm (1.0–4.2 cm)	Colorectal 25/25 (100%)	TACE + RFA	48.4 months	None
Wang et al. [18]	3.5 ± 1.3 cm	Breast 88/88 (100%)	TACE + RFA	15.6 months	1 bone marrow suppression, 1 infection (non-specified), 1 severe digestive tract symptoms, and 1 liver and kidney damage
Alexander, E.S. et al. [19]	4.7 cm (1.5–8.0 cm)	Colorectal 18/42 (43%) Sarcoma 5/42 (12%) Cholangiocarcinoma 5/42 (12%) Pancreatic neuroendocrine 4/42 (10%) Pancreatic adenocarcinoma 4/42 (10%) Breast 3/42 (7%) Appendiceal 1/42 (2%) Esophagus b1/42 (2%) Pyriform squamous 1/42 (2%)	TACE + RFA (32) or MWA (8) or CA (2)	55% (95% CI 40–71%) at 1 year and 30% (95% CI 16–45%) at 2 years	3 liver abscesses, 2 groin bleeds, 1 pseudoaneurysm, 1 portal vein thrombus with lobar infarct, 1 retroperitoneal hematoma, and 1 biliary fistula
Faiella et al. [20]	3.7 cm (3.2–7.3 cm)	Breast 10/22 (45%) Colorectal 10/22 (45%) Neuroendocrine tumors 2/22 (5%) Leiomyosarcoma 2/22 (5%)	TAE + MWA	No residue/recurrence detected at the CT scan follow-up (2 years)	1 bleeding
Kobe et al. [21]	3.6 ± 0.6 cm (range: 3–5 cm)	Colorectal 8/46 (17%) Sarcoma 10/46 (22%) Adrenal carcinoma 10/46 (22%) Thyroid carcinoma 3/46 (7%) Neuroendocrine tumor 4/46 (9%) Breast 3/46 (7%) Other 8/46 (17%)	TACE + RFA (34) or MWA (11) or CA (1)	95% at 1 year and 77% at 2 years	1 pleural effusion, 1 segmental, 1 portal vein thrombosis, and 1 subcapsular hematoma

**Table 3 jcm-11-05576-t003:** TAE/TACE methods of the studies included in the review.

Authors	TAE/TACE Methods
Gadaleta et al. [13]	23 patients: 35 mg epirubicin plus 15 mg mitomycin C 9 patients: 100 mg doxorubicin loaded on Dc-Beads (Biocompatibles, Farnham, UK) 1 patient: 100 mg doxorubicin loaded on Hepasphere (BioSphere Medical, Roissy, Cedex, France) 4 patients: 100 mg irinotecan loaded on Dc-Beads
Fong, Z.V. et al. [14]	50 mg cisplatin, 50 mg adriamycin, and 10 mg mitomycin emulsified with Lipiodol (Guerbet, Bloomington, IN, USA) and Gelfoam slurry
Kan et al. [15]	20–50 mg adriamycin or 4–10 mg mitomycin C emulsified with 2 to 12.5 mL Lipiodol UltraFluid, followed by embolization with gelatin sponge particles
Wu et al. [16]	50–150 mg oxaliplatin, 10–50 mg epirubicin, and 1.5–10 mL Lipiodol, followed by embolization with gelatin sponge particles for insufficient embolization cases
Yamakado et al. [17]	A total of 2–6 mg mitomycin C was dissolved in 5 mL of distilled water and mixed with 300 mg degradable starch microspheres (Spherex; Yakult Co., Ltd., Tokyo, Japan).
Wang et al. [18]	Vinorelbine + capecitabine with or without trastuzumab, docetaxel + capecitabine with or without trastuzumab
Alexander, E.S. et al. [19]	Prior to 2010: 100 mg cisplatin, 50 mg doxorubicin, and 10 mg mitomycin C, followed by 150–250 µm Contour PVA Embolization Particle (Boston Scientific, Marlborough, MA, USA) From 2010 on: 50 mg doxorubicin, and 10 mg mitomycin C, followed by 100–300 µm Embospheres (Merit Medical, Salt Lake City, UT, USA)
Faiella et al. [20]	Embozene Microspheres 75 µm and 100 µm spheres (Embozene, Color-Advanced Microspheres; Celonova BioSciences, Peachtree City, GA, USA)
Kobe et al. [21]	Doxorubicin or irinotecan mixed with Lipiodol (32 lesions) (followed by embolization with gelatin sponge) or drug-eluting beads (Boston Scientific) (14 lesions)

## Data Availability

The data and the code used for the experiments are available on request.

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
