# Peer review of "Combined Trans-Arterial Embolization and Ablation for the Treatment of Large (>3 cm) Liver Metastases: Review of the Literature"

_jcm, 2022, doi:10.3390/jcm11195576_

Round 1

Reviewer 1 Report

Overall: Nice review of the available literature for combined embolization and ablation in patients with secondary liver metastasis of lesion that are greater than 3 cm. There is a fair amount of heterogeneity in the selected studies which may make it difficult to extract data and have meaningful results.  If English is not the native language of the authors, I would suggest an English Editorial Service to review the manuscript for grammar and style. 

Title: Combined trans arterial embolization and ablation for the treatment of large (> 3 cm) liver metastases: review of the literature 

Good. Clear.

Abstract: Concise and understandable. line 27. delete the extra "in"

Introduction: Nice introduction and easy to read/follow. 
line 62, delete "the" prior to complete. line 63, replace "similarly" with similar. 

Materials and Methods:  Good methodology. However, there is at least one newer study that I can think of that may have been missed:

Kobe A, Tselikas L, et al. Single-session transarterial chemoembolization combined with percutaneous thermal ablation in liver metastases 3 cm or larger. Diagnostic and Interventional imaging. June 2022. 

The heterogeneity of the data makes it challenging to make sense of the data but the authors should be commended for attempting it.

Discussion:  Heterogeneity of data, small sample sizes, and other limitations to draw meaningful conclusion - which the authors did a decent job of highlighting. This combined approach could be considered safe with low complications rates - but need more details. The authors only glance over it instead of giving us numbers/evidence. Need more data/evidence for overall survival and for progression-free survival extracted from the studies you analyzed. The last paragraph is not necessarily needed. It is anecdotal and not necessarily evidence-based; furthermore, the reference cited is for HCC not secondary liver cancer. It is out of place in this context. Would suggest removing it. 

Conclusion:  Fine. 

Figures and Tables:

Fig 1: Good

Table 1: Good

Fig 2: Nice images (only need one to illustrate the metastatic lesion). Do need arrows/arrowheads to point out the findings and reference it in your text. 

Figure 3: The images are not very clear and there is no labeling. Need to label the individual pictures (A, B, C, D, etc). Need arrows/arrowheads/etc to label the relevant findings. Need to reference this in your text so that the readers can follow. Need to specify the company and the state/country if you are referencing a commercial product - EMBOZENE. 

Figure 4. Nice images and follow-up with labeling of the individual pictures (a, b, c). But, still needs arrows/arrowheads pointing to the relevant findings and a description in the text.

Author Response

Reviewer 1
Overall: Nice review of the available literature for combined embolization and ablation in patients with secondary liver metastasis of lesion that are greater than 3 cm. There is a fair amount of heterogeneity in the selected studies which may make it difficult to extract data and have meaningful results.  If English is not the native language of the authors, I would suggest an English Editorial Service to review the manuscript for grammar and style. 
Thanks to the reviewer for the comment. An English revision has been made.
Title: Combined trans arterial embolization and ablation for the treatment of large (> 3 cm) liver metastases: review of the literature 
Good. Clear.
Abstract: Concise and understandable. line 27. delete the extra "in"
The second “in” has been removed
Introduction: Nice introduction and easy to read/follow. 
line 62, delete "the" prior to complete. line 63, replace "similarly" with similar. 
Thanks for the suggestions. The modifies have been made
Materials and Methods:  Good methodology. However, there is at least one newer study that I can think of that may have been missed:
Kobe A, Tselikas L, et al. Single-session transarterial chemoembolization combined with percutaneous thermal ablation in liver metastases 3 cm or larger. Diagnostic and Interventional imaging. June 2022. 
The reference has been added. We apologize, this reference is more recent than the first draft of the manuscript.
The heterogeneity of the data makes it challenging to make sense of the data but the authors should be commended for attempting it.
Discussion:  Heterogeneity of data, small sample sizes, and other limitations to draw meaningful conclusion - which the authors did a decent job of highlighting. This combined approach could be considered safe with low complications rates - but need more details. The authors only glance over it instead of giving us numbers/evidence. Need more data/evidence for overall survival and for progression-free survival extracted from the studies you analyzed. The last paragraph is not necessarily needed. It is anecdotal and not necessarily evidence-based; furthermore, the reference cited is for HCC not secondary liver cancer. It is out of place in this context. Would suggest removing it. 
Thanks to the reviewer for these comments. We reported evidence for overall survival and for progression-free survival in the Discussion part (“The overall survival was reported by only 4 works. However there is a wide range among studies: while Wu et al [16] and Alexander et al [19] reported a survival rate of 46.7% and 55% at 12 months respectively, the other two works showed very high rates (89.4% and 88%) [15, 17]. The two works which analysed the overall survival comparing the group treated with the combined approach and a control group, reported in both cases better results for the observation group [14, 18]. The data are also few about progression-free survival the data, with only two works reporting an absolute value.
We may conclude that the treatment with TACE + ablative techniques is an emerging and promising approach and this review demonstrates the effectiveness and efficacy on a small set of samples. At the moment, there are no long-term results and prospective studies are next to come.”) and removed the last paragraph as suggested.
Conclusion:  Fine. 
Figures and Tables:
Fig 1: Good
Table 1: Good
Fig 2: Nice images (only need one to illustrate the metastatic lesion). Do need arrows/arrowheads to point out the findings and reference it in your text. 
Figure 3: The images are not very clear and there is no labeling. Need to label the individual pictures (A, B, C, D, etc). Need arrows/arrowheads/etc to label the relevant findings. Need to reference this in your text so that the readers can follow. Need to specify the company and the state/country if you are referencing a commercial product - EMBOZENE. 
Figure 4. Nice images and follow-up with labeling of the individual pictures (a, b, c). But, still needs arrows/arrowheads pointing to the relevant findings and a description in the text.
All the reviewer’ suggested modifies have been made about Figures

Reviewer 2 Report

The authors have written an article about the efficacy and use of combining ablation and embolization in patients with secondary liver lesions. Eight articles are included and described in the final review and the authors conclude that the combined par roach of ablation and embolisation in liver lesions greater than 3 cm is safe and has positive effects on patient survival. The article I well written and the subject is interesting. There are however a few issues that needs to be addressed. 

The treatment is, as described, considered in patients with unresectable metastases yet it is unclear whether the aim is to convert to respectability or if this is to be considered as a solely palliative therapy in the included articles. If the combined procedure is part of a conversion strategy or palliative therapy other therapies ought to be discussed or included in the comparison before concluding its efficacy. 

The conclusion that the combined procedure is safe and efficient in lesions greater than 3 cm needs to be clarified. According to the manuscript it is not clear if all  included articles included a cut-off of 3cm. 

There is no statistical evaluation included or mentioned. Even though the number of included articles are few this ought to evaluated and/or discussed/motivated. 

Author Response

Dear reviewer,
first of all, thank you for your comments on our article.
We did not include a discussion regarding the reason for the combined treatment, whether palliative or as a bridge to surgical treatment, because the individual articles do not address the topic, considering only overall post-treatment survival. 
We clarified in the text that the articles included in the review studied liver lesions, including lesions larger than 3 cm. Although the mean size of the treated lesions in this review is greater than 3 cm, and in some articles 3 cm  was considered as a cut off, we preferred not to exclude articles that did not reach this cut off in order to not further reduce the number of studies concerning an already niche topic.
We preferred not to perform a statistical analysis both because of the small number of articles, and of patients, but especially because of the considerable variability in both study methods and study presentation regarding end points and statistical analysis.

Reviewer 3 Report

The authors are trying to report the usefulness of combinatorial therapy of TAE and ablation for the metastatic lesions in the liver with greater than 3 cm. The authors reviewed total of eight reviews focusing on the safety and therapeutic efficacy. While the topic might be interesting, there are several concerns to be addressed prior to the further review.

Firstly, while the authors are trying to focus the safety and efficacy of the combinatorial therapy, the information from the only eight papers are not clearly demonstrated. The authors may want to include the tables showing the detailed information. In this table, authors need to show the number or percentage of each metastatic tumor, as the efficacy of TAE is dependent on the tumors. (for e.g., metastasis form the colon (n=xx, yy%), breast cancer (n=zz, aa%)) 

Secondary, as the safety assessment (data collection) is relatively poor, they need to revise the description.

In addition, the authors may want to update the references, as there are a few reviews useful (for e.g., PMID: 35748208, PMID: 35049694, PMID: 34914005).

Author Response

Dear reviewer,
first of all, thank you for your comments on our article.
We have expanded the information in the eight articles with another table (Table 2), which adds information about the individual histotypes of the primary tumors responsible for the liver lesions, as well as information regarding the outcomes (including median survival) and safety (number of major complications).

Regarding safety, we added a sentence to the discussion on the complication rates that are low and not different from those of the single thermo and embo / chemo methods, citing the reference standards “Complication rates were no different than those of individual ablation and embolization methods, as defined by the CIRSE Standards of Practice on Hepatic Transarterial Chemoembolisation and Thermal Ablation [22, 23].”

Thank you for your valuable suggestions regarding the bibliography, which has been updated.

Round 2

Reviewer 2 Report

If/when a statistical analysis cannot be performed it ought to be mentioned as a limitation.

Author Response

Thanks to the reviewer for this comment. We performed a descriptive analysis, inserted two new tables and we added a sentence in the last paragraph of the discussion as suggested (limitations).

Reviewer 3 Report

The authors have revised the manuscript appropriately, therefore, can be considered for its acceptance.

Author Response

Thanks to the Reviewer for this opportunity and for al  the suggestions in round 1 
